# Laughing over a Drink: Exploring the Relationship Between Humor Styles and Drinking Behavior

**DOI:** 10.3390/bs15111580

**Published:** 2025-11-18

**Authors:** Giulia Baldacci, Angelo Marino, Lia Cirillo, Lucia Guidi, Alessandra Musio, Alberto Dionigi

**Affiliations:** 1Studi Cognitivi, 47921 Rimini, Italy; 2IRCCS Istituto delle Scienze Neurologiche di Bologna, 40139 Bologna, Italy; 3Centro Hercolani, 40124 Bologna, Italy; 4Noesis Clinical Center, 47841 Cattolica, Italy

**Keywords:** humor, humor styles, alcohol, coping strategies, individual differences

## Abstract

Humor is a key component of social relationships and has been linked to both positive health outcomes and detrimental effects, depending on the style of humor employed. However, its associations with alcohol-related behaviors remain largely underexplored. The present study investigated the relationships between humor styles, gender, age, and alcohol use. A total of 392 adults (123 males and 269 females), aged between 18 and 74 years (*M* = 36.64; *SD* = 13.11), completed the Humor Styles Questionnaire (HSQ). This data was used to assess humor styles and the Alcohol Use Disorders Identification Test (AUDIT) to evaluate alcohol consumption patterns. Results indicated that male sex and younger age were significantly associated with higher alcohol use. Moreover, Self-Defeating humor emerged as a significant positive predictor of alcohol-related behavior. These findings suggest that certain maladaptive humor styles, particularly Self-Defeating humor, may play a role in reinforcing problematic drinking behaviors, possibly by serving as a dysfunctional coping mechanism in social contexts.

## 1. Introduction

Humor is a universal human experience, and over the past four decades, research has focused on its fundamental role in people’s lives, with particular attention to its contribution to mental health ([28]; [38]). Humor represents a multifaceted construct that integrates emotional, cognitive, and behavioral components ([28]; [36]). Specifically, its potential to enhance well-being is expressed through three interconnected dimensions: (a) a cognitive component, namely the recognition of a stimulus as humorous; (b) an affective response, characterized by feelings of mirth; and (c) a physiological reaction, such as smiling or laughter ([36]). Moreover, humor has been shown to facilitate psychological distancing from difficulties, thereby fostering positive emotions and reducing tension and anxiety, rendering it an effective coping strategy in stressful situations ([1]; [43]). Humor has been associated with many positive outcomes and represents a fundamental aspect of psychological life, contributing to the promotion of a healthy lifestyle ([27]). In recent decades, scientific interest in the role of humor in health promotion has seen considerable growth, with researchers examining its relationship with both interpersonal and intrapersonal well-being ([32]). However, the relationship between humor and alcohol-related behavior remains largely unexplored. Previous studies found that both extraversion and alcohol consumption influence facial and verbal expressions of humor-related enjoyment ([35]). Extraverts generally displayed more frequent and intense smiles and laughter, but their positive reactions decreased under high alcohol use, suggesting a moderating effect of alcohol on their emotional expression. In contrast, a low dose of alcohol enhanced facial expression and enjoyment in extroverts, while introverts’ responses remained largely unaffected. Overall, the findings indicate that personality traits and alcohol interact to influence humor-induced positive effects, with effects mediated by individual differences in positive emotionality. More recently, it was found that alcohol consumption increased smiling expression and positive emotional expression, but not in direct response to the humorous moments of the comedy routine. Instead, alcohol enhanced smiling during non-humorous or neutral periods, suggesting that its social effects arise less from amplifying reactions to external stimuli and more from enhancing the general enjoyment of company. These findings indicate that alcohol primarily strengthens social bonding and positive effects in otherwise less stimulating social moments ([11]). Research emphasizes that humorous alcohol advertisements can stimulate interpersonal communication and shape alcohol-related attitudes ([20]).

Although there is considerable evidence of the relationship between humor and various aspects of everyday life, the link between humor and alcohol-related behavior remains almost completely unexplored. The present study was designed as a preliminary investigation to enrich the scientific literature and explore the relationship between alcohol-related behavior in the general population—without necessarily implying a categorical diagnosis of Alcohol Use Disorder—and humor styles. To broaden the theoretical understanding of this relationship, we propose a preliminary conceptual framework linking humor styles and drinking behavior. Specifically, self-defeating humor may influence alcohol use through two complementary psychological pathways. First, an intrapersonal mechanism based on maladaptive emotion regulation, in which alcohol consumption serves as a short-term coping strategy to alleviate negative effect reinforced by self-defeating humor. Second, an interpersonal mechanism, whereby alcohol facilitates social bonding or masks insecurity stemming from low self-esteem. These mechanisms are consistent with emotion regulation and coping theories ([28]; [43]) and may clarify why self-defeating humor is associated with problematic drinking patterns. This model provides a theoretical foundation for the empirical findings reported in the present study.

### 1.1. Humor Styles

Numerous attempts have been made to provide an exhaustive and representative account of humor depending on various theoretical frameworks ([28]). Martin and colleagues ([29]) contributed to this debate by introducing the concept of specific humor styles, defined as expressions of personality traits that are consistently related to both personality dimensions and psychological well-being. Among the four styles proposed, two are considered adaptive or “benevolent” (Affiliative humor and Self-enhancing humor), while the other two are classified as maladaptive or “non-benevolent” (Aggressive humor and Self-defeating humor). Affiliative humor reflects the tendency to share jokes and amusing remarks with others, using humor to strengthen social bonds and create a positive atmosphere. Self-enhancing humor indicates the inclination to maintain a humorous outlook on life and to rely on humor as a coping strategy in the event of stress or adversity. Aggressive humor involves the use of sarcasm, ridicule, or disparaging remarks, often at the expense of others, without regard for the potential negative impact on the target. Self-defeating humor is characterized by self-disparagement, where individuals make themselves the object of jokes to entertain others and gain social approval. Humor styles characterized by disparaging or avoidant tendencies, such as the Aggressive and Self-defeating styles, may have detrimental effects on individual well-being. Conversely, humor styles associated with positive psychological functioning include the Affiliative style, which serves to foster and maintain social relationships, and the Self-enhancing style, which promotes a self-ironic yet strengthening perspective on oneself ([29]).

Regarding gender differences in humor, a comprehensive meta-analysis of studies published between 1977 and 2018 found that men tend to score higher in Aggressive humor ([22]). Findings on humor production, however, remain inconsistent: while some studies reported no significant differences, others suggested that men are perceived as funnier, particularly when using critical or aggressive forms of humor ([22]; [25]). From an evolutionary perspective, gender differences in humor can be understood through the lens of sexual selection and reproductive strategies. Humor may be appealing as a signal of genetic quality, a concept that traces back to Darwin’s theory of sexual selection ([8]). According to this framework, humor evolved, partly as a signal of genetic fitness, intelligence, and creativity, traits advantageous in mate selection ([31]). Within this context, women became more selective in choosing partners, favoring men who displayed indicators of high genetic quality. Humor production, particularly witty or intelligent humor, may have served as one such indicator. Men, in contrast, are thought to have evolved to use humor as a courtship strategy to attract potential mates and signal mental fitness ([17]).

Social factors further shape these evolutionary tendencies: gender roles and cultural expectations reinforce the pattern of men as humor producers and women as appreciative audiences ([16]). In many societies, men are socially encouraged to be funny as a demonstration of confidence and superior status; on the contrary, women’s humor is often undervalued or judged by different standards. Thus, the interplay between evolutionary foundations and social norms helps explain both the biological origins and contemporary expressions of gender differences in humor. From an evolutionary and developmental perspective, humor styles change across the lifespan as individuals’ social and emotional goals evolve. Younger people tend to use humor to establish social bonds and gain group acceptance, whereas with age, humor increasingly serves adaptive functions related to emotion regulation and psychological well-being. Age-related patterns also emerged: adolescents and young adults, especially those of university age, tend to rely more on Affiliative humor, adults typically make use of both Affiliative and Self-enhancing humor, whereas older adults show a greater tendency to adopt Self-enhancing humor ([46]).

#### Alcohol Consumption and Its Relationship with Humor

Alcohol consumption is widespread across all ages and cultures. The longstanding production and consumption of alcohol has led it to be regarded as “the best known and most widely used means of altering human consciousness” ([26]). According to the World Health Organization ([47]), in 2019, the average global consumption of alcohol was about 5.5 L per person aged 15 years and older per year. In addition to the overall level of consumption, the way people drink is also crucial for health outcomes ([21]). Moderate drinking has been investigated in relation to several major diseases common in Western societies. For instance, some studies suggest that the risk of developing various health problems rises by about 5% for every 10 g of alcohol consumed per day ([4]). In the psychological domain, Alcohol Use Disorder (AUD) is the most recent diagnostic term used to describe a heterogeneous spectrum of clinical manifestations. Historically, alcohol abuse and alcohol dependence were considered distinct conditions; however, the fifth edition of the Diagnostic and Statistical Manual of Mental Disorders (DSM-5) eliminated this distinction, defining AUD as a continuum of severity characterized by a cluster of behavioral and physiological symptoms ([3]). The eleven criteria for AUD can be grouped into four domains: impaired control, social impairment, risky use, and pharmacological indicators such as tolerance and withdrawal. A diagnosis of AUD requires the presence of at least two criteria, with the total number of criteria endorsed determining the severity of the disorder.

The treatment of alcohol associated behavior represents a critical and complex phase of the therapeutic process, requiring a bio-psycho-social approach that integrates medical, psychological, and social perspectives ([7]). The intensity and type of intervention vary according to the severity of alcohol use. Among psychological treatments, Motivational Interviewing (MI) and Motivational Enhancement Therapy (MET) have demonstrated effectiveness in resolving ambivalence and increasing readiness to change, particularly in individuals with less severe dependence ([44]). Cognitive Behavioral Therapy (CBT) remains one of the most effective and widely validated approaches, as it provides patients with strategies to identify high-risk situations, challenge dysfunctional beliefs, strengthen coping skills, and prevent relapses. In addition, 12-step programs are frequently incorporated into stepped-care frameworks ([18]). Patient motivation is the cornerstone of recovery, and tailoring treatment intensity to individual needs is essential.

To the best of our knowledge, only two studies have explored, in different ways, the connection between alcohol-related behavior and humor. One research study integrated Laughter Therapy into a group treatment for patients with substance use disorders and reported improvements in self-esteem, happiness, and anxiety among participants ([9]). Another study investigated the relationship between alcohol dependence and humor styles in a large sample of 2752 adults ([41]). Participants completed the Humor Styles Questionnaire to assess their humor styles and took part in a telephone survey to evaluate their alcohol use and related problems. The survey identified individuals who satisfied the DSM-IV criteria for Alcohol Dependence ([2]). Results indicated that male gender and Aggressive humor were significant predictors of alcohol dependence ([41]).

The present study was designed as a preliminary investigation to enrich scientific literature and explore the relationship between alcohol-related behavior in the general population, without necessarily implying a categorical diagnosis of Alcohol Use Disorder, and humor styles. This approach also aims to provide a foundation for future studies involving clinical samples. Based on existing literature, the study hypotheses were as follows: we expected younger individuals and males to report higher alcohol consumption. Regarding humor styles, we expect greater alcohol use among individuals who predominantly employ non-benevolent humor styles. Other variables were examined in an exploratory manner.

## 2. Method

### 2.1. Procedure

The study adopted a cross-sectional design. Data was collected through an online survey distributed via social media platforms and mailing lists. Participants were members of the general population, and participation was voluntary. Prior to participation, respondents were presented with an informed consent form outlining the objectives of the study and the voluntary nature of participation. The research was conducted in compliance with the ethical standards of the Noesis Clinical Center Ethics Committee (Approval Code: 001EP/2025) and with the principles of the Declaration of Helsinki. Anonymity and confidentiality of the data were guaranteed to all participants. Inclusion criteria required individuals to be Italian citizens and at least 18 years of age. Participants were recruited using the snowball sampling method ([15]), with invitations to participate in the study. The survey was created and administered through the REDCap platform, which ensures anonymity ([19]). Participants were required to complete the full set of questionnaires. The latter included measures of sociodemographic characteristics (age, gender, marital status, and educational level) and questionnaires assessing humor and alcohol consumption.

### 2.2. Partecipants

A total of 392 adults (123 males and 269 females), aged between 18 and 74 years (*M* = 36.64; *SD* = 13.11), completed the surveys. Participants were mostly adults with varying levels of education (1.8% had completed middle school, 26.5% held a high school diploma, 19.1% held a bachelor’s degree, 31.4% held a master’s degree, and 21.2% had a level of education higher than a master’s degree).

### 2.3. Measures

Humor Styles Questionnaire (HSQ) ([29]) is an instrument designed to measure the use of different humor styles. The questionnaire consists of 32 items, divided into four subscales of 8 items each, assessing humor styles that can be grouped into two categories: two adaptive or benevolent styles (Affiliative and Self-Enhancing humor) and two maladaptive or non-benevolent styles (Aggressive and Self-Defeating humor). Examples of items are presented as follows. Affiliative humor: *“I enjoy making people laugh.”* Self-Enhancing humor: *“I don’t need other people to make me laugh—I can usually find something funny even when I’m by myself.”* Aggressive: *“If someone makes a mistake, I will often tease them about it.”* Self-Defeating humor: *“I let people laugh at me or make fun at my expense more than I should.”* Participants were asked to indicate their level of agreement with each statement using a 4-point Likert scale (1 = strongly disagree; 4 = strongly agree). In the present study, we used the Italian version translated and validated by Penzo and colleagues ([33]). The four scales employed in this study showed satisfactory reliability (Cronbach’s α: Affiliative = 0.82; Self-Enhancing = 0.75; Aggressive = 0.72; Self-Defeating = 0.75).

Alcohol Use Disorders Identification Test (AUDIT) ([40]). The AUDIT is a questionnaire developed by the World Health Organization (WHO) to effectively and reliably identify alcohol-related disorders in the adult population. It consists of 10 items designed to assess the quantity and frequency of alcohol consumption, its negative consequences, and related symptoms. Each question provides closed-ended responses scored from 0 to 4, yielding a total score ranging from 0 to 40, with higher scores indicating greater alcohol use. Unlike other screening tools that mainly identify severe disorders, the AUDIT allows the detection of both alcohol use disorders and hazardous drinking, and it is also useful in identifying key symptoms of alcohol dependence and specific consequences of excessive drinking. It has been widely validated across different cultural and clinical contexts, showing high reliability in assessing drinking behavior (α = 0.80). In the present study, the AUDIT was chosen because it measures alcohol consumption without providing a formal diagnosis of alcoholism, making it particularly suitable for a sample drawn from the general population rather than from a clinical setting. We employed the validated Italian version ([34]).

### 2.4. Statistical Analysis

All statistical analyses were carried out using SPSS version 25.0 ([23]). Prior to conducting the main analyses, a priori power analysis was performed to determine the required sample size, employing G*Power software 3.1.9.7 ([13], [12]). Power analysis is a fundamental component of research design, as it enables the estimation of the minimum number of participants needed to detect effects of a given magnitude at a predetermined level of confidence. For the present study, the analysis indicated that, in the case of a linear multiple regression, a minimum of 218 participants would ensure a statistical power of 0.90, assuming a very small effect size (f^2^ = 0.05) and an α level of 0.05. Recent guidelines on the interpretation of Pearson’s correlation effect sizes in the field of individual differences research recommend considering correlations of 0.10, 0.20, and 0.30 as relatively small, typical, and relatively large, respectively ([14]). Descriptive statistics (means, standard deviations, skewness, and kurtosis) were computed for all study variables to assess central tendency, variability, and distributional properties. Skewness and kurtosis values were examined to evaluate normality; values between −1 and +1 were considered acceptable. Bivariate relationships among gender, age, HSQ, and AUDIT scores were examined through Pearson’s correlation coefficients. Subsequently, hierarchical multiple regression analyses were conducted with AUDIT as the dependent variable and the four humor styles as predictors. Multicollinearity diagnostics were examined and confirmed to be within acceptable thresholds prior to model estimation. Cases with missing data were excluded listwise.

## 3. Results

### 3.1. Intercorrelations Among Variables

Correlations among gender, age, humor styles, and alcohol behavior are reported in Table 1.

Overall, alcohol consumption levels in the sample were relatively low, a finding that is consistent with expectations for a non-clinical population. When examining gender differences, female participants reported significantly lower scores on Affiliate and Aggressive humor compared to their male counterparts, as well as lower levels of alcohol-related behavior. Furthermore, alcohol-related behavior was positively associated with male sex and with younger age, suggesting that men and younger individuals in the sample were more likely to engage in risky drinking patterns. Regarding humor styles, alcohol consumption showed significant positive correlations with Affiliative, Aggressive, and Self-Defeating humor. This indicates that individuals who frequently engage in social and light-hearted humor, but also those who employ humor in more hostile or self-disparaging ways, tend to report higher levels of alcohol use.

### 3.2. Humor Styles as Predictors of Alcohol-Related Behavior

Subsequently, a hierarchical linear regression was conducted (Table 2). The dependent variable was the score obtained on the AUDIT scale. In the first step, gender and age were entered as independent variables, while in the second step, the scales measuring the use of different humor styles (Affiliative, Self-enhancing, Aggressive, and Self-defeating) were included.

The results of the regression analysis are reported in Table 2 and illustrate the progression across the different steps of the analysis. In Step 1, the predictors of alcohol-related behavior were male gender (*β* = −0.22; *p* < 0.001) and younger age (*β* = −0.20; *p* < 0.001), accounting for 8% of the explained variance (F(2, 389) = 18.511, *p* < 0.001). In Step 2, when humor styles were added to the model, the Self-defeating humor style emerged as a significant predictor of alcohol-related behavior (*β* = 0.17, *p* < 0.01), indicating a positive correlation between this humor style and alcohol consumption. The model accounted for 21% of the variance.

## 4. Discussion

The purpose of the present research was to investigate the relationships between humor styles and alcohol use, with the broader intention to investigate previously underexplored associations. By adopting an individual-differences perspective, this study examined how distinct humor styles predict alcohol-related behavior, thus deepening the theoretical understanding of humor as a psychological coping mechanism. Integrating the Humor Styles framework ([29]) with models of emotion regulation and self-presentation enables a more systematic interpretation of the findings and their implications for drinking behavior. First, descriptive analyses indicated that the distributions of the humor style variables were approximately normal, with skewness and kurtosis values within the acceptable range. This suggests that participants’ responses were well distributed across the humor style dimensions. Conversely, age and alcohol use scores (AUDIT) showed positive skewness and higher kurtosis, reflecting a predominance of younger participants and a concentration of low alcohol use levels in the sample. These distributional characteristics are consistent with expectations for community-based samples and support the reliability of the observed associations among humor styles and alcohol consumption. Significant gender differences in alcohol consumption were observed. Consistent with previous research, men reported higher levels of alcohol use compared to women ([24]). Gender differences also emerged in humor styles: females exhibited lower levels of Aggressive and Affiliative humor than males. This finding is in line with prior studies showing that women tend to score lower on Aggressive humor ([22]). [10] ([10]) suggested that, among men, aggressive humor may be associated with a greater perception of receiving social support, which could partially explain its higher prevalence in male participants.

When examining the predictors of alcohol-related behavior, the present results were similar to established findings, confirming both younger age and male gender as significant predictors of alcohol use. These results highlight the relevance of sociodemographic factors in shaping drinking behaviors, and they emphasize the importance of integrating individual difference variables, such as humor styles, into the broader framework of alcohol research. From an evolutionary perspective, humor can be understood as a social and mating strategy, functioning as a signal of intelligence, creativity, and social competence. In this context, the higher levels of humor use and alcohol consumption typically observed among men may reflect behaviors aimed at enhancing social bonding and mate attraction: functions that are particularly relevant in younger individuals. As individuals grow older, their social and personal goals related to humor may transform, which can lead to differences in how they use humor and in their drinking behaviors. These considerations suggest that humor-related individual differences may not only shape social interactions but also influence alcohol use patterns within an evolutionary framework ([25]). The main purpose was to capture both the behavioral and demographic aspects of alcohol consumption and the psychological mechanisms, such as humor styles, which are more likely to increase problematic use or, conversely, help reduce it.

Regarding the specific correlations between alcohol-related behavior and humor styles, a positive association emerged with self-defeating humor. This style is characterized by the tendency to ridicule oneself. This humor style involves making oneself the target of jokes to gain social acceptance or to ease interpersonal tension. Individuals who adopt it often exhibit low self-esteem, heightened anxiety and sadness, and dissatisfaction with their relationships. Moreover, self-defeating humor has been associated with a range of negative psychological outcomes, including depressive symptoms and poor emotional well-being ([29]), and it has been linked to poorer social skills, particularly a reduced ability to assert one’s rights, to act assertively, and to accurately understand others’ emotions ([48]). In line with the findings reported by Cann and colleagues ([6]), self-defeating humor has also been identified as a significant predictor of perceived stress.

This tendency may be particularly relevant for those with alcohol related problems, as alcohol is often used as a coping strategy to manage negative effects and interpersonal insecurity. Smucker Barnwell and colleagues ([45]) suggested that heavy drinkers tend to use maladaptive emotion regulation strategies, which align with the use of self-defeating humor to cope with social or emotional discomfort. Moreover, Maurage and colleagues ([30]) found that individuals with alcohol use disorders exhibit emotional dysregulation and altered self-perception, which could significantly contribute to the tendency to devalue oneself through humor. It is therefore plausible to assume that, in the presence of lower social competence, higher levels of perceived stress, and greater negative affectivity, the consumption of alcoholic beverages may serve as a dysfunctional coping strategy to regulate emotions. Individuals who frequently use self-defeating humor may rely on alcohol as an additional means to manage emotional discomfort or enhance social belonging, reinforcing maladaptive behavioral patterns. Over time, as alcohol use turns chronic, such self-deprecating tendencies might reinforce a negative self-concept and perpetuate the cycle of alcohol misuse. Furthermore, alcohol may be used in accordance with one of the functions of self-defeating humor, namely, as a strategy for regulating or masking one’s genuine emotional state in order to appear more socially acceptable, even at the cost of personal well-being. In contrast, adaptive humor styles, such as affiliative and self-enhancing humor, may play a protective role by facilitating emotional regulation, fostering positive affect and promoting supportive social interactions ([5]). These mechanisms highlight the importance of considering humor style theory within broader models of drinking behavior, particularly in the context of prevention and intervention strategies, by showing that both emotional dysregulation and social self-esteem concerns may mediate the relationship between self-defeating humor and alcohol use, as suggested by the proposed dual-pathway model.

The present findings differ from previous evidence showing that aggressive humor was the main predictor of alcohol dependence ([41]). In our study, which focused on alcohol-related behaviors rather than clinical dependence, self-defeating humor emerged as the strongest predictor. A possible explanation lies in the different psychological dynamics involved in alcohol use versus alcohol dependence. While aggressive humor reflects interpersonal hostility and maladaptive relational patterns that are more likely to characterize individuals with a consolidated dependence, self-defeating humor is associated with low self-esteem, social anxiety, and a tendency to seek approval by ridiculing oneself. Taken together, these results suggest that self-defeating humor may play a relevant role in the early stages of problematic drinking, whereas aggressive humor could eventually turn more prominent in the progression toward alcohol dependence.

Since humor is closely related to emotion regulation and coping strategies ([28]), individuals who adopt self-enhancing humor are likely to possess more effective tools to cope with stress without resorting to alcohol. Conversely, self-defeating humor may reinforce problematic drinking patterns, especially among individuals prone to emotional dysregulation. Research has also focused on programs designed to enhance humor skills, expression, and appreciation. For instance, [38] ([38]) have developed interventions aimed at fostering adaptive humor styles and improving emotional well-being through structured humor training. Research demonstrated that the 7 Humor Habits Program effectively enhances participants’ sense of humor, as perceived by both themselves and their peers, while also increasing cheerful mood, reducing seriousness, and improving overall life satisfaction ([39]). A possible avenue of application lies in the context of alcohol-related therapies, where promoting positive social emotions and adaptive humor use may contribute to healthier coping strategies and more rewarding interpersonal interactions. From an applied perspective, these results also point toward possible clinical and preventive applications. Humor-based interventions for alcohol use could be designed to help individuals recognize and modify self-defeating humor tendencies. Structured programs such as the 7 Humor Habits Program ([39]) could be adapted to integrate humor-skills training with motivational or cognitive-behavioral techniques. For instance, exercises promoting self-enhancing humor may improve emotion regulation and reduce the reliance on alcohol as a maladaptive coping strategy. Developing such tailored interventions may enhance both emotional well-being and social connectedness in individuals at risk of problematic drinking.

Such combined approaches may promote positive emotions, strengthen social coping mechanisms, and support healthier behavioral patterns, ultimately contributing to more effective prevention and treatment outcomes. Moreover, fostering awareness of self-defeating humor and promoting its constructive modification could further enhance these positive effects. Future research should therefore investigate the potentially protective role of adaptive humor styles in preventing alcohol misuse, with particular attention to their interplay with metacognitive strategies.

It is important to acknowledge several limitations of the present study. First, given the cross-sectional design, causal inferences cannot be made, and caution is therefore required when interpreting the directionality of the results. Future research should employ alternative research designs to confirm and extend our findings. Longitudinal or experimental approaches would allow for a more precise examination of the directional relationships between humor styles and drinking behavior. Another limitation concerns the assessment of humor, which was measured exclusively through the humor styles model. Although the HSQ is a widely used and validated tool, some studies have noted limitations in its structure and cross-cultural adaptations ([37]; [42]). Future research could consider alternative measures, such as Ruch’s Comic Styles approach, to obtain more generalizable and theoretically nuanced insights into humor-related individual differences.

The study relied exclusively on self-report measures, which may limit the validity of the findings. Future research should include additional methods, such as peer or observational assessments, and replicate the study with samples of individuals who have experienced drinking problems to enhance generalizability. Further, the sample consisted mainly of female participants and individuals with higher education levels, which may reduce the generalizability of the results to the broader population. Specifically, the high proportion of female and highly educated participants may limit the representativeness of the sample. Future studies should replicate these findings in more diverse populations, including participants of different socioeconomic backgrounds and cultural contexts, to enhance external validity. Moreover, the study did not control for potential confounding variables such as depression, anxiety, or personality traits, which are known to influence both humor use and alcohol consumption. Future research should include these variables to test the robustness and specificity of the observed associations.

Despite these limitations, the present study highlights that humor, depending on how it is expressed and which style is employed, may exert some influence on alcohol-related behaviors. Understanding these interactions is crucial, as certain humor styles may either foster or mitigate problematic drinking patterns. Moreover, these findings may help develop targeted interventions that use humor to support mental health and lower the risks linked to alcohol consumption. Overall, the present findings emphasize the importance of considering humor as a meaningful psychosocial factor in understanding alcohol-related behaviors. Future studies could build on this work by testing the effectiveness of humor-based interventions in promoting healthier coping and social interaction patterns.

## Figures and Tables

**Table 1 behavsci-15-01580-t001:** Correlations among variables.

	M	SD	Skeweness	Curtosis	Gender	Age	Affiliative	Self Enhancing	Aggressive	Self Defeating
Age	36.64	13.11	1.05	0.12	−0.01					
HSQ_Affiliative	25.26	3.66	−0.39	0.17	−0.10 *	−0.25 **				
HSQ_Self_enhancing	21.15	3.3	−0.18	0.18	−0.06	0.09	0.27 **			
HSQ_Aggressive	16.41	2.75	0.17	0.24	−0.24 *	−0.05	0.14 **	0.13 **		
HSQ_Self_defeating	19.23	3.42	0.03	0.11	−0.03	−0.13 *	0.25 **	0.26 **	0.36 **	
AUDIT	4.30	3.90	1.96	5.00	−0.20 **	−0.20 **	0.17 *	−0.01	0.21 **	0.22 **

*Note. N* = 392; Male = 1; Female = 2; AUDIT = Alcoholic Behavior. * *p* < 0.05, ** *p* < 0.01.

**Table 2 behavsci-15-01580-t002:** Predictors of Alcohol Use (AUDIT Scores).

	AUDIT	
Predictors	ΔR^2^	*Sig.*
Step 1	0.08 ***	
Gender		−0.22 ***
Age		−0.20 ***
Step 2	0.13 ***	
Affiliative Humor		0.08
Self-Enhancing Humor		−0.08
Aggressive Humor		0.10
Self-Defeating Humor		0.17 **
Total R^2^	21%	

*Note. N* = 392; Male = 1; Female = 2; AUDIT = Alcoholic Behavior. ** *p* < 0.01, *** *p* < 0.001.

## Data Availability

The dataset supporting this study is publicly available via Mendeley Data (Dionigi, 2025): https://doi.org/10.17632/x7xxsbkdpg.1.

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
