# Peer review of "Laughing over a Drink: Exploring the Relationship Between Humor Styles and Drinking Behavior"

_behavsci, 2025, doi:10.3390/bs15111580_

Round 1
Reviewer 1 Report
Comments and Suggestions for Authors
Scientific Repports
Reviewer for „Laughing over a Drink: Exploring the Relationship Between Two Humor Styles and Drinking Behavior“
Thank you for the opportunity to review the manuscript "Laughing over a Drink: Exploring the Relationship Between Two Humor Styles and Drinking Behavior." The manuscript aims to contribute to filling the gap in knowledge regarding the relationship between drinking behavior and humor styles. The authors designed a precise cross-sectional study to provide additional information intended to offer substantial evidence on the examined issues and applicable findings. I believe that the study may offer a solid contribution to the investigated area; however, at the same time, there are several aspects that should be further discussed. There are, in particular, several important issues that require serious revision before recommending the manuscript for publication.
Introduction
The content of the Introduction section generally presents adequate and relevant findings that provide insight into the existing framework of evidence regarding the relationship between humor and alcohol. However, I believe that additional evidence should be incorporated. Although alcohol consumption is analyzed in relation to humor styles, presenting findings on the relationship between alcohol consumption and other aspects or models of humor would be highly beneficial. For example, illustrating relationships between alcohol consumption and dimensions from other models (such as Ruch’s) or other aspects of humor (beyond everyday humor use) could be valuable. Including such findings would provide a more comprehensive understanding of the relationship between the examined constructs and offer a stronger basis for more generalizable conclusions.
It seems to me that the Introduction section would benefit from a clearer structure. In its current form, information is presented without a discernible logical flow (at least to me), and I would suggest the following reorganization:
To include subsections within the Introduction section.
Begin with a brief introductory part including the opening text (Lines 25–36), and then add the following sentences: “The present study was designed as a preliminary investigation to enrich the scientific literature and explore the relationship between alcohol-related behavior in the general population—without necessarily implying a categorical diagnosis of Alcohol Use Disorder—and humor styles. This approach also aims to provide a foundation for future studies involving clinical samples.”
Include a subsection titled “Humor Styles,” which would incorporate the relevant parts of the Introduction (Lines 79–96).
Include a subsection titled “Alcohol Consumption and Its Relationship with Humor,” which would encompass the following lines: 37–66 and 97–114, as well as the hypothesis section: “Based on existing literature, the study hypotheses were as follows: we expected younger individuals and males to report higher alcohol consumption. Regarding humor styles, we expected greater alcohol use among individuals who predominantly employ non-benevolent humor styles. Other variables were examined exploratorily.”
I truly believe that such structuring would provide a clearer, step-by-step introduction to the study topic. I emphasize that it is up to the authors to decide whether to adopt this suggestion, but I encourage them to consider the potential benefits that such reorganization could bring to their manuscript.
The authors could provide more detailed information on differences in humor styles between men and women, including evolutionary foundations and how these may interact with social factors.
The authors may also include information on programs designed to improve humor skills and expression (or reception), such as those developed by Willibald Ruch. This addition would nicely complement the discussion of therapies related to alcohol-related issues.
Method
The sample should be described in more detail. How were the participants recruited and assessed—online or using a paper-and-pencil procedure? Was participation voluntary?
In the Introduction section, it is stated that the study was conducted on a general population sample; this should be clarified and supported in the Methods section.
Results
Adjust Tables 1 and 2 according to APA 7th edition guidelines (only basic horizontal lines should be used).
Since the mean (M) values are already reported in Table 2, it is unnecessary to repeat these values in parentheses within the text when interpreting results.
Explanations for asterisks (p-values) should be included in the Notes section.
Skewness and kurtosis values should be included in Table 1 and discussed in the Results section to provide a more precise understanding of the respondents’ score distributions.
Discussion
The authors’ explanation of sex differences and their consistency with previous results is noted.
In addition, the authors write: "When examining the predictors of alcohol-related behavior, the present results were similar to established findings, confirming both younger age and male gender as significant predictors of alcohol use. These results highlight the relevance of sociodemographic factors in shaping drinking behaviors, and they emphasize the importance of integrating individual difference variables, such as humor styles, into the broader framework of alcohol research." I would like to hear what these findings would mean from the perspective of evolutionary explanations provided by recent studies. As humor changes across the lifespan, and since men generally score higher, how could this be interpreted from an evolutionary standpoint (in terms of the goals for which humor is typically used)? Furthermore, how can these findings be applied in practical interventions? The authors mention both “treatments/workshops” related to humor and treatments related to alcohol. It would be valuable to elaborate on how these specific findings could be integrated into a broader theoretical and practical framework. That would represent a meaningful contribution to the study whose “purpose was to investigate the relationships between humor styles and alcohol use, with the broader intention to investigate previously underexplored associations,” and would emphasize the true novelty of the present work.
In line with this, could the authors propose more direct suggestions regarding the potential combination of certain interventions or therapeutic procedures (related to alcohol-related problems) with programs such as Ruch’s, which aim to improve humor use and skills? I would particularly appreciate such elaboration for two reasons: 1) it would constitute a direct implication that extends beyond the current study, and 2) it would, in general, significantly increase the quality and informativeness of the manuscript.
Conclusion
The conclusion is generally adequate; however, I believe that with the revisions suggested for the Discussion section, the authors might find it appropriate to add one or two sentences to further support and emphasize the overall findings. It is, of course, up to them to decide whether they wish to do so.
Limitations and future directions
It would be beneficial to replicate the study using samples consisting of individuals who have or have had drinking problems, in order to provide more generalizable conclusions.
The lack of multiple measurement methods is a limitation. The conclusions were based exclusively on self-assessments. It would be advisable to include additional measures, such as peer assessments or observational data, in future studies.
I hope the authors will not mind me raising so many concerns. The study is good but has a great deal of unrealized potential. I truly believe that the study could make a solid contribution to the examined topic if the manuscript is improved in line with the suggestions provided. I wish the authors all the best and hope that my comments will motivate them to make additional efforts to strengthen the manuscript.rove the manuscript.
Author Response
Reviewer #1
Thank you for the opportunity to review the manuscript "Laughing over a Drink: Exploring the Relationship Between Two Humor Styles and Drinking Behavior." The manuscript aims to contribute to filling the gap in knowledge regarding the relationship between drinking behavior and humor styles. The authors designed a precise cross-sectional study to provide additional information intended to offer substantial evidence on the examined issues and applicable findings. I believe that the study may offer a solid contribution to the investigated area; however, at the same time, there are several aspects that should be further discussed. There are, in particular, several important issues that require serious revision before recommending the manuscript for publication.
Response: Thank you very much for your faithful comments to our study and the provided insights. We have dealt at our best with them in order to improve the clarity of the paper in accordance to your suggestions.
Introduction
The content of the Introduction section generally presents adequate and relevant findings that provide insight into the existing framework of evidence regarding the relationship between humor and alcohol. However, I believe that additional evidence should be incorporated. Although alcohol consumption is analyzed in relation to humor styles, presenting findings on the relationship between alcohol consumption and other aspects or models of humor would be highly beneficial. For example, illustrating relationships between alcohol consumption and dimensions from other models (such as Ruch’s) or other aspects of humor (beyond everyday humor use) could be valuable. Including such findings would provide a more comprehensive understanding of the relationship between the examined constructs and offer a stronger basis for more generalizable conclusions.
Response: Thank you very much for pointing out this issue. We have incorporated your suggestion and included a general discussion of studies examining the influence of alcohol on smiling behavior and within social contexts. However, no studies were found addressing the connection between different models of humor styles (specifically, the comic styles markers developed by Ruch and colleagues) and alcohol use.
It seems to me that the Introduction section would benefit from a clearer structure. In its current form, information is presented without a discernible logical flow (at least to me), and I would suggest the following reorganization:
To include subsections within the Introduction section. Begin with a brief introductory part including the opening text (Lines 25–36) and then add the following sentences: “The present study was designed as a preliminary investigation to enrich the scientific literature and explore the relationship between alcohol-related behavior in the general population—without necessarily implying a categorical diagnosis of Alcohol Use Disorder—and humor styles. This approach also aims to provide a foundation for future studies involving clinical samples.”
Include a subsection titled “Humor Styles,” which would incorporate the relevant parts of the Introduction (Lines 79–96).
Include a subsection titled “Alcohol Consumption and Its Relationship with Humor,” which would encompass the following lines: 37–66 and 97–114, as well as the hypothesis section: “Based on existing literature, the study hypotheses were as follows: we expected younger individuals and males to report higher alcohol consumption. Regarding humor styles, we expected greater alcohol use among individuals who predominantly employ non-benevolent humor styles. Other variables were examined exploratorily.”\
I truly believe that such structuring would provide a clearer, step-by-step introduction to the study topic. I emphasize that it is up to the authors to decide whether to adopt this suggestion, but I encourage them to consider the potential benefits that such reorganization could bring to their manuscript.
Response: Thank you very much for your detailed suggestion, which we have incorporated, as we believe it has improved the fluency and clarity of the paper.
The authors could provide more detailed information on differences in humor styles between men and women, including evolutionary foundations and how these may interact with social factors.
Response: Thank you very much for the helpful suggestion. We have now added a clarifying paragraph to address this point.
The authors may also include information on programs designed to improve humor skills and expression (or reception), such as those developed by Willibald Ruch. This addition would nicely complement the discussion of therapies related to alcohol-related issues.
Response: We appreciate the reviewer’s valuable suggestion. Accordingly, we have added information regarding the programs utilized by Willibald Ruch and Paul McGhee.
Method
The sample should be described in more detail. How were the participants recruited and assessed—online or using a paper-and-pencil procedure? Was participation voluntary? In the Introduction section, it is stated that the study was conducted on a general population sample; this should be clarified and supported in the Methods section.
Response: We would like to thank the reviewer for this observation. This information was already included in the manuscript: “The study adopted a cross-sectional design. Data were collected through an online survey distributed via social media platforms and mailing lists…” “Participants were recruited using the snowball sampling method [24], with invitations to take part in the study. The survey was created and administered through the REDCap platform, which ensures anonymity [25].”
Nevertheless, to provide further clarification, we have added the following sentence: “Participants were members of the general population, and participation was voluntary.”
Results
Adjust Tables 1 and 2 according to APA 7th edition guidelines (only basic horizontal lines should be used).
Response: Done.
Since the mean (M) values are already reported in Table 2, it is unnecessary to repeat these values in parentheses within the text when interpreting results.
Response: We thank the reviewer for this comment. We were uncertain about this point, as the means are reported in Table 1 (and not in Table 2). However, since the means are already presented in Table 1, we have removed their repetition from the text following the table.
Explanations for asterisks (p-values) should be included in the Notes section.
Response: Correct. We apologize for this omission and have now addressed it in the revised version of the manuscript.
Skewness and kurtosis values should be included in Table 1 and discussed in the Results section to provide a more precise understanding of the respondents’ score distributions.
Response: We thank the reviewer for this helpful suggestion. Skewness and kurtosis values have been calculated and included in Table 1, and their interpretation has been added to the Discussion section.
Discussion
The authors’ explanation of sex differences and their consistency with previous results is noted. In addition, the authors write: "When examining the predictors of alcohol-related behavior, the present results were similar to established findings, confirming both younger age and male gender as significant predictors of alcohol use. These results highlight the relevance of sociodemographic factors in shaping drinking behaviors, and they emphasize the importance of integrating individual difference variables, such as humor styles, into the broader framework of alcohol research." I would like to hear what these findings would mean from the perspective of evolutionary explanations provided by recent studies. As humor changes across the lifespan, and since men generally score higher, how could this be interpreted from an evolutionary standpoint (in terms of the goals for which humor is typically used)?
Response: Thank you very much for this thoughtful and stimulating comment. We fully agree that considering the findings from an evolutionary perspective adds valuable depth to the interpretation. Accordingly, we have added a paragraph in the Discussion section elaborating on how humor and alcohol-related behaviors may be understood within an evolutionary framework, particularly in relation to social bonding and mating functions, as well as age-related changes in these dynamics.
Furthermore, how can these findings be applied in practical interventions? The authors mention both “treatments/workshops” related to humor and treatments related to alcohol. It would be valuable to elaborate on how these specific findings could be integrated into a broader theoretical and practical framework. That would represent a meaningful contribution to the study whose “purpose was to investigate the relationships between humor styles and alcohol use, with the broader intention to investigate previously underexplored associations,” and would emphasize the true novelty of the present work.
Response: We agree with the reviewer, and we have therefore added a brief section in the Discussion. Specifically, we mention that interventions aimed at enhancing adaptive humor styles (e.g., Ruch & McGhee, 2014; Ruch et al., 2018) could represent a promising avenue in alcohol-related prevention and therapy, by fostering positive emotions and healthier coping strategies.
In line with this, could the authors propose more direct suggestions regarding the potential combination of certain interventions or therapeutic procedures (related to alcohol-related problems) with programs such as Ruch’s, which aim to improve humor use and skills? I would particularly appreciate such elaboration for two reasons: 1) it would constitute a direct implication that extends beyond the current study, and 2) it would, in general, significantly increase the quality and informativeness of the manuscript.
Response: Thank you, we have therefore added a brief section in the Discussion.
Conclusion
The conclusion is generally adequate; however, I believe that with the revisions suggested for the Discussion section, the authors might find it appropriate to add one or two sentences to further support and emphasize the overall findings. It is, of course, up to them to decide whether they wish to do so.
Response: We thank the reviewer for this helpful suggestion. We have followed the recommendation and added a couple of sentences at the end of the Conclusion section to further emphasize the main findings and their broader implications.
Limitations and future directions
It would be beneficial to replicate the study using samples consisting of individuals who have or have had drinking problems, in order to provide more generalizable conclusions. The lack of multiple measurement methods is a limitation. The conclusions were based exclusively on self-assessments. It would be advisable to include additional measures, such as peer assessments or observational data, in future studies.
Response: We have added these two limitations in the related section.
I hope the authors will not mind me raising so many concerns. The study is good but has a great deal of unrealized potential. I truly believe that the study could make a solid contribution to the examined topic if the manuscript is improved in line with the suggestions provided. I wish the authors all the best and hope that my comments will motivate them to make additional efforts to strengthen the manuscript.rove the manuscript.
Response: We sincerely thank the reviewer for their thoughtful feedback and encouraging words. We truly appreciate the constructive comments and have carefully revised the manuscript in line with the suggestions provided. We are confident that these revisions have helped to strengthen the clarity, depth, and overall contribution of the study.
Reviewer 2 Report
Comments and Suggestions for Authors
Comments to the Author
Dear author(s),
This study aims to explore the relationship between humor styles and drinking behavior, demonstrating both theoretical value and practical significance. The research design is sound, the data analysis methods are generally appropriate, and the conclusions provide valuable insights for mental health interventions. However, there are certain limitations in terms of theoretical innovation, methodological rigor, and interpretation of results.
Firstly, regarding theoretical contribution and innovation, this study introduces humor style theory into the field of drinking behavior research, addressing a gap in the existing literature. The finding that “self-defeating humor” predicts drinking behavior is particularly insightful. Nevertheless, the study essentially applies an existing theoretical model without delving deeply into the underlying mechanisms through which humor styles influence drinking behavior. The theoretical framework is relatively underdeveloped and fails to integrate effectively with mainstream theories of drinking behavior. It is recommended that the discussion provide a more in-depth analysis of the mechanisms underlying the role of “self-defeating humor” and strengthen the exploration of the potential protective effects of adaptive humor styles.
Secondly, in terms of research methodology, the study features an adequate sample size, passes statistical power tests, employs measurement tools with good reliability and validity, and applies statistical methods appropriately. However, the sample is characterized by a relatively high proportion of female participants and generally higher education levels, which may limit the representativeness of the findings. The cross-sectional design prevents causal inference, and potential confounding variables such as anxiety and depression were not controlled for. Additionally, the exclusive reliance on self-report data introduces the risk of common method bias. It is recommended that the discussion fully address these methodological limitations and specify the methods used to handle missing data.
Finally, regarding the paper’s logic and structure, the overall framework is complete, the literature review is comprehensive, and the results are clearly presented. However, the introduction is somewhat lengthy and could be more focused on the theoretical connection between humor styles and drinking behavior. The interpretation of results in the discussion section appears somewhat speculative and lacks sufficient empirical support. The conclusions are relatively general and fail to propose specific intervention recommendations or clear directions for future research. It is advised to streamline the introduction, strengthen the empirical foundation of the discussion, and conclude with more actionable practical applications.
Best regards.
Author Response
Reviewer #2
This study aims to explore the relationship between humor styles and drinking behavior, demonstrating both theoretical value and practical significance. The research design is sound, the data analysis methods are generally appropriate, and the conclusions provide valuable insights for mental health interventions. However, there are certain limitations in terms of theoretical innovation, methodological rigor, and interpretation of results.
Response: We thank the reviewer for the appreciation of our study. We have now revised the manuscript according to your valuable comments, as detailed below.
Firstly, regarding theoretical contribution and innovation, this study introduces humor style theory into the field of drinking behavior research, addressing a gap in the existing literature. The finding that “self-defeating humor” predicts drinking behavior is particularly insightful. Nevertheless, the study essentially applies an existing theoretical model without delving deeply into the underlying mechanisms through which humor styles influence drinking behavior. The theoretical framework is relatively underdeveloped and fails to integrate effectively with mainstream theories of drinking behavior. It is recommended that the discussion provide a more in-depth analysis of the mechanisms underlying the role of “self-defeating humor” and strengthen the exploration of the potential protective effects of adaptive humor styles.
Response: Thank you very much for highlighting this important point, which also reflects Reviewer 1’s comments. We have now expanded the discussion on the relationship between self-defeating humor style and alcohol-related behavior, incorporating references to recent and relevant studies to provide a deeper theoretical grounding. Additionally, we have elaborated on the potential mechanisms linking humor styles to drinking behavior and discussed the possible protective effects of adaptive humor styles.
Secondly, in terms of research methodology, the study features an adequate sample size, passes statistical power tests, employs measurement tools with good reliability and validity, and applies statistical methods appropriately. However, the sample is characterized by a relatively high proportion of female participants and generally higher education levels, which may limit the representativeness of the findings. The cross-sectional design prevents causal inference, and potential confounding variables such as anxiety and depression were not controlled for. Additionally, the exclusive reliance on self-report data introduces the risk of common method bias. It is recommended that the discussion fully address these methodological limitations and specify the methods used to handle missing data.
Response: We thank the reviewer for this helpful comment. A specific limitation regarding gender and age has been added to the Limitations section, along with references to the correlational design of the study, the potential influence of confounding variables, and the reliance on self-report data. Missing data were handled using listwise deletion, including only participants with complete data on all study variables. A corresponding sentence has also been added to the Data Analyses section.
Finally, regarding the paper’s logic and structure, the overall framework is complete, the literature review is comprehensive, and the results are clearly presented. However, the introduction is somewhat lengthy and could be more focused on the theoretical connection between humor styles and drinking behavior. The interpretation of results in the discussion section appears somewhat speculative and lacks sufficient empirical support. The conclusions are relatively general and fail to propose specific intervention recommendations or clear directions for future research. It is advised to streamline the introduction, strengthen the empirical foundation of the discussion, and conclude with more actionable practical applications.
Response: We sincerely thank the reviewer for these insightful and constructive comments. These suggestions were integrated with those provided by the first reviewer, leading us to substantially revise several sections of the manuscript, including both the Introduction and Discussion. We have made these parts more focused and detailed, strengthened the theoretical and empirical foundations, and expanded the bibliography to better support our arguments and conclusions.
Best regards.
Reviewer 3 Report
Comments and Suggestions for Authors
This study uses a sample recruited cumulatively which is not representative of the whole population (it is highly educated, for instance). The use of the full range of the AUDIT means that it is measuring general level of involvement with alcohol rather than either alcohol use disorder or frequency and level of drinking. It seemed to me the paper might recognise the interpersonal nature of much drinking, and that humorous conversation is a substantial aspect of social worlds of heavy drinking. The styles of humour measured by the four types would interact with the kinds of humour associated with sitting around together and drinking. Qualitative research on social worlds of heavy drinking might be drawn on to make sense of the findings on the relation of AUDIT score with the four types of humour.
All in all, I found this to be a study from which I learned rather little, and with a sample which meant I didn't trust what I learned anyway. It's relatively well written, but I don't think it contributes much.
Author Response
Reviewer #3
This study uses a sample recruited cumulatively which is not representative of the whole population (it is highly educated, for instance). The use of the full range of the AUDIT means that it is measuring general level of involvement with alcohol rather than either alcohol use disorder or frequency and level of drinking. It seemed to me the paper might recognise the interpersonal nature of much drinking, and that humorous conversation is a substantial aspect of social worlds of heavy drinking. The styles of humour measured by the four types would interact with the kinds of humour associated with sitting around together and drinking. Qualitative research on social worlds of heavy drinking might be drawn on to make sense of the findings on the relation of AUDIT score with the four types of humour.
All in all, I found this to be a study from which I learned rather little, and with a sample which meant I didn't trust what I learned anyway. It's relatively well written, but I don't think it contributes much.
Response: We thank the reviewer for this detailed and thoughtful feedback. We acknowledge the limitation related to the sample characteristics, which were not fully representative of the general population due to the relatively high education level of participants. This limitation has been explicitly stated and discussed in the revised version of the manuscript.
Regarding the use of the AUDIT scale, we agree that it captures general levels of alcohol involvement rather than specific patterns of alcohol use disorder or drinking frequency. This point has been clarified in the Methods and Discussion sections.
We also appreciate the reviewer’s valuable insight on the interpersonal and social dimensions of drinking behavior and the potential role of humor within these contexts. Following this suggestion, we have expanded the Discussion to consider the social and conversational aspects of humor and drinking, emphasizing how different humor styles might interact with the social nature of alcohol consumption.
Finally, we have revised the manuscript extensively to enhance its theoretical depth, strengthen the interpretation of findings, and improve the overall contribution to the literature on humor and drinking behavior.
Round 2
Reviewer 1 Report
Comments and Suggestions for Authors
Please find the attachment

Author Response
We would like to sincerely thank the reviewer for the thoughtful and constructive feedback, as well as for the positive assessment of our revised manuscript. We are pleased that the revisions were found satisfactory and have carefully addressed the two additional suggestions provided.
Comment:
The first suggestion concerns the relationship between age and humor styles. A few sentences explaining the general relation
between humor styles and age from an evolutionary perspective would be desirable (in the same way that the authors
expanded the explanation of the gender–humor styles relationship). This is not mandatory, but it would make the text more
compatible with the previous paragraph and provide a more comprehensive insight into this relationship.
Response:
We thank the reviewer for this insightful observation. We have added a short introductory explanation about the relationship between humor styles and age from an evolutionary and developmental perspective at the beginning of the relevant paragraph in the Introduction section. This addition provides a clearer theoretical rationale for age-related differences in humor use.
Comment:
The second suggestion relates to the limitations of the study. I am not sure if this was included as a comment in the first round of the review, so I am raising this point now. Although the HSQ is the most popular and widely used measure of humor styles, studies have consistently reported certain shortcomings of this instrument. The works of Ruch and Heintz, as well as Silvia and Rodriguez (2020), have raised serious concerns based on their findings. Moreover, translations of the HSQ into different languages have not always been conducted adequately, and certain inconsistencies can be observed when comparing HSQ content across countries. Therefore, it may be useful to mention that alternative measures, such as Ruch’s Comic Styles approach, could be considered in future studies to obtain more generalizable conclusions. Having this in mind, I would suggest that the authors include this point as a limitation of the present study.
Response:
We appreciate this valuable suggestion. We have expanded the Limitations section to acknowledge the potential shortcomings of the HSQ and to note that alternative approaches, such as Ruch’s Comic Styles framework, may provide more generalizable insights in future research.
We have incorporated both suggestions into the revised manuscript, which we believe further improves its theoretical completeness and overall clarity. We are very grateful for the reviewer’s careful reading and constructive feedback.
Reviewer 2 Report
Comments and Suggestions for Authors
Comments to the Author
Dear author(s),
It is encouraging to see that you have addressed the previously raised issues in detail. However, several points still require further revision.
Firstly, it is recommended to strengthen the elaboration of the theoretical mechanism. The current discussion on the psychological pathways linking self-defeating humor and drinking behavior remains preliminary. We suggest introducing a concise theoretical model in the introduction or discussion section to clarify the potential mediating or moderating roles of these variables, thereby enhancing the theoretical depth of the study.
Secondly, regarding the methodology section, three key points need attention. First, the sample limitations should be explicitly acknowledged. The current sample, characterized by a high proportion of female participants and a generally higher education level, may limit the generalizability of the findings. It is recommended to clearly state this limitation in the discussion and suggest that future research validate these results in more diverse populations. Second, clarification on variable control is necessary. The study did not account for potential confounding variables such as depression, anxiety, or personality traits. This should be explicitly noted in the “Limitations” section, with a recommendation for future studies to include these variables to strengthen the robustness of the conclusions. Third, the limitations regarding causal inference must be clarified. Given the cross-sectional design, causal inferences cannot be made. This should be explicitly stated in the discussion, accompanied by a suggestion for future research to employ longitudinal or experimental designs for further validation.
Finally, concerning the overall logic, the discussion section could be further integrated with theory. First, the current discussion is somewhat descriptive; it is advisable to interpret the findings more systematically within a theoretical framework. Second, the practical implications section could be more specific. While it mentions the potential application of “humor interventions”in alcohol treatment, it lacks concrete intervention strategies or implementation pathways. We recommend proposing more actionable suggestions..
Best regards.
Author Response
Reviewer’s comment:
“It is recommended to strengthen the elaboration of the theoretical mechanism. The current discussion on the psychological pathways linking self-defeating humor and drinking behavior remains preliminary. We suggest introducing a concise theoretical model in the introduction or discussion section to clarify the potential mediating or moderating roles of these variables, thereby enhancing the theoretical depth of the study.”
Response: We fully agree with the reviewer’s observation. In response, we have added a concise theoretical model describing two complementary psychological pathways: an intrapersonal mechanism (maladaptive emotion regulation) and an interpersonal mechanism (social self-esteem regulation). In this way self-defeating humor may influence alcohol use. This model is presented at the end of the Introduction and referenced again in the Discussion to interpret the findings.
“The sample limitations should be explicitly acknowledged. … Clarification on variable control is necessary … The limitations regarding causal inference must be clarified.”
Response: We appreciate these important points and have revised the Limitations section accordingly to explicitly address each one.
- Sample limitations:
We now acknowledge that the overrepresentation of female and highly educated participants may restrict generalizability.
- Control variables:
We have added a clear note that potential confounding variables (e.g., depression, anxiety, personality traits) were not controlled for. Moreover, the study did not control for potential confounding variables such as depression, anxiety, or personality traits, which are known to influence both humor use and alcohol consumption. Future research should include these variables to test the robustness and specificity of the observed associations.
- Causal inference:
We explicitly note the limitations of our cross-sectional design and recommend longitudinal or experimental approaches.
Reviewer’s comment:
“The discussion section could be further integrated with theory. The current discussion is somewhat descriptive; it is advisable to interpret the findings more systematically within a theoretical framework.”
Response: We have strengthened the theoretical integration by explicitly connecting the empirical findings with the dual-pathway model introduced in the Introduction.
Reviewer’s comment:
“The practical implications section could be more specific. While it mentions the potential application of ‘humor interventions’ in alcohol treatment, it lacks concrete intervention strategies or implementation pathways.”
Response: We agree that the practical implications required further development. We have therefore expanded the final section of the Discussion to include specific, actionable intervention strategies based on established humor-training programs.